# Bioprocessing of Two Crop Residues for Animal Feeding into a High-Yield Lovastatin Feed Supplement

**DOI:** 10.3390/ani12192697

**Published:** 2022-10-07

**Authors:** Amaury Ábrego-García, Héctor M. Poggi-Varaldo, M. Teresa Ponce-Noyola, Graciano Calva-Calva, Cutberto José Juvencio Galíndez-Mayer, Gustavo G. Medina-Mendoza, Noemí F. Rinderknecht-Seijas

**Affiliations:** 1Department of Biotechnology and Bioengineering, CINVESTAV-IPN, P.O. Box 14-740, Mexico City 07000, Mexico; 2Environmental Biotechnology and Renewable Energies Group, CINVESTAV-IPN, P.O. Box 14-740, Mexico City 07000, Mexico; 3Departamento de Ingeniería Bioquímica, ENCB, Instituto Politécnico Nacional, Av. Wilfrido Massieu 399, Nueva Industrial Vallejo, Mexico City 07738, Mexico; 4Escuela Superior de Ingeniería Química e Industrias Extractivas, Instituto Politécnico Nacional, Av. Luis Enrique Erro S/N, Nueva Industrial Vallejo, Gustavo A. Madero, Mexico City 07738, Mexico

**Keywords:** *Aspergillus terreus*, crop residues, lovastatin, solid-state fermentation

## Abstract

**Simple Summary:**

Lovastatin is a fungal secondary metabolite that can mitigate rumen methane production. This work aimed at evaluating the lovastatin production by solid-state fermentation from selected crop residues and *A. terreus* strains, considering the post-fermented residues as feed supplements for ruminants. Fermented oat straw by *A. terreus* CDBB H-194 exhibited the highest lovastatin yield (23.8 mg/g DM fed). GC–MS analysis identified only a couple of compounds from the residues fermented by CDBB H-194 (1,3-dipalmitin trimethylsilyl ether in the fermented oat straw) and stearic acid hydrazide in the fermented wheat bran) that could negatively affect ruminal bacteria and fungi.

**Abstract:**

This work aimed to evaluate the lovastatin (Lv) production by solid-state fermentation (SSF) from selected crop residues, considering the post-fermented residues as feed supplements for ruminants. The SSF was performed with two substrates (wheat bran and oat straw) and two *A. terreus* strains (CDBB H-194 and CDBB H-1976). The Lv yield, proximate analysis, and organic compounds by GC–MS in the post-fermented residues were assessed. The combination of the CDBB H-194 strain with oat straw at 16 d of incubation time showed the highest Lv yield (23.8 mg/g DM fed) and the corresponding degradation efficiency of hemicellulose + cellulose was low to moderate (24.1%). The other three treatments showed final Lv concentrations in decreasing order of 9.1, 6.8, and 5.67 mg/g DM fed for the oat straw + CDBB H-1976, wheat bran + CDBB H-194, and wheat bran + CDBB H-1976, respectively. An analysis of variance of the 2^2^ factorial experiment of Lv showed a strong significant interaction between the strain and substrate factors. The kinetic of Lv production adequately fitted a zero-order model in the four treatments. GC–MS analysis identified only a couple of compounds from the residues fermented by *A. terreus* CDBB H-194 (1,3-dipalmitin trimethylsilyl ether in the fermented oat straw and stearic acid hydrazide in the fermented wheat bran) that could negatively affect ruminal bacteria and fungi. Solid-state fermentation of oat straw with CDBB H-194 deserves further investigation due to its high yield of Lv; low dietary proportions of this post-fermented oat straw could be used as an Lv-carrier supplement for rumen methane mitigation.

## 1. Introduction

Oat straw and wheat bran are common feeds for ruminants. They are characterized by moderate to high lignocellulose content and low digestibility [1]. Therefore, different pre-treatment processes improve their nutritional value and digestibility. In this way, solid-state fermentation (SSF) has enhanced rumen digestibility through enzymatic delignification of high-fiber forages [2].

An important bioproduct obtained from fungal SSF is lovastatin (Lv). The latter is a significant drug in treating hypercholesterolemia [3]. Another essential feature of the Lv is that it can mitigate rumen methane emissions [4]. Specifically, Lv disrupts the cell membrane synthesis of methanogenic archaea and inhibits the growth of methanogens [5,6].

Nevertheless, the pure Lv utilization as an anti-methanogenic additive for livestock is limited because of its high cost [5]. As shown in Table 1, many options are available for the Lv production. The SSF bioprocess might provide the least-cost option [7].

The advantages of SSF mentioned above have attracted the attention of researchers on Lv production. In the period 2011–2021, up to seven reviews on the production of Lv by SSF with agricultural residues or feedstocks were published [15,16,17,18,19,20,21]. We attempted to characterize the literature contributions in terms of several criteria based on this literature search. An ad hoc score criterion was also proposed: 0 (absent) or 1 (present), the importance of the criteria is discussed in Appendix A. The list of the criteria follows: Sa, strain availability; Pas, proximate analysis of substrates; Des, degradation efficiency of substrates; Sar, statistical analysis of results; Klp, the kinetics of Lv production; Coc, characterization of organic compounds from post-fermented substrates.

Interestingly, this translates into an average score of 3.2/6 of the relevant literature. This score confirms that the scope of previous research is limited, and there is room for improving the research on Lv production by SSF of agricultural residues, mainly when these are destined for animal production. For instance, the characterization of organic compounds from post-fermented substrates is essential because some metabolites can benefit livestock [4]. However, others such as mycotoxins might cause important health and performance problems [22,23]. Overall, most studies on the fermented substrates as actual or potential dietary supplements to mitigate rumen methane emissions have focused on the final yield of Lv [4,12,24,25,26].

Thus, this work aimed to evaluate the Lv production by SSF from selected agricultural residues, considering the nutrient assessment and the characterization of organic compounds from post-fermented substrates. For this purpose, two substrates (wheat bran and oat straw) and two *A. terreus* strains (CDBB H-194 and CDBB H-1976) were assessed.

## 2. Materials and Methods

### 2.1. Chemical Analysis of Crop Residues

The substrates oat straw (*Avena sativa*) and wheat bran (*Triticum aestivum*) were dried at 50 °C for 48 h and kept in zip lock bags at 4 °C. Proximate analysis was determined as described by the AOAC methods [27]: Dry matter 934.01 (DM), ether extract 920.29 (EE); ash 942.05, and crude protein 976.05 (CP). Cell wall variables: neutral detergent fiber (NDF), acid detergent fiber (ADF), hemicellulose, cellulose, and lignin were evaluated according to Van Soest et al. [28], and a fiber analyzer was used (Ankom Technology, model no. A-200, Macedon, NY, USA), and NDF was carried out with sodium sulfite and stable amylase.

### 2.2. Inoculum and Culture Propagation

*Aspergillus terreus* CDBB H-194 and CDBB H-1976 were obtained from the National Collection of Microbial Strains of CINVESTAV-IPN, Mexico. The strains were kept at 4 °C on potato dextrose agar medium afterwards; they were sub-cultured every 30 d. Fungi spores were scrapped and suspended in a solution of Tween 80 in sterile deionized water (0.1%, *v*/*v*). Spores were counted with a hemocytometer and adjusted to 1 × 10^7^ spores/mL. Finally, 2 mL of each suspension was utilized as the inoculum [29].

### 2.3. Conditions of Solid-State Fermentation

Five grams of the substrate (either oat straw or wheat bran) was ground (5 mm) and transferred to a 250 mL Erlenmeyer flask. The moisture content was adjusted to 70% with a liquid medium (concentrations in g/L, unless otherwise stated): CaCl_2_, 0.3; KH_2_PO_4_, 2.1; ZnSO_4_, 0.3; MgSO_4_, 0.3; NaNO_3_, 0.5; methionine, 1.4, and glycerol, 20 mL/L [30]. The culture medium pH was set to 5.5 with 1 M H_2_SO_4_. Then, the flasks were sterilized at 121 °C for 15 min, cooled to room temperature, and seeded with 2 mL of spore suspension (CDBB H-194 or CDBB H-1976). Abiotic controls were carried out with both inactive *Aspergillus* strains. All experiments were performed in triplicate. The cultures were kept at 30 °C for 72 h, followed by 24 h at 28 °C. Afterwards, the temperature was kept at 26 °C [31]. The flasks were shaken twice a day. Sampling was performed at 0, 6, 12, and 16 d.

### 2.4. Determination of Lovastatin

Post-fermented substrates (oat straw and wheat bran) were dried in a forced-air convection oven (55 °C for 24 h) and powdered using a mortar and pestle. The powdered samples (1 g) were mixed with 40 mL of ethyl ethanoate in Erlenmeyer’s flasks and agitated for 5 h at 200 rpm. The mixture was filtered through a membrane filter of 0.22 μm (Durapore, Millipore, MA, USA). The ethyl ethanoate was eliminated in a Rotavapor Model R3 (Büchi Labortechnik, AG, Switzerland). The dry residue was dissolved in 5 mL of cyanomethane and filtered through 0.22 μm acrodisc syringe filters (Millipore, MA, USA) for further analysis [32].

Lovastatin was determined as reported elsewhere [33] using an HPLC-UV (Varian Analytical Instruments, Model 9010, Palo Alto, CA, USA) with a Gemini C18 column (Phenomenex, Torrance, CA, USA) and acetonitrile: H_2_O (70:30, *v*/*v*) as a mobile phase containing H_3_PO_4_ at 0.1% (*v*/*v*) (0.5 mL/min flow rate). The wavelength of detection was 237 nm. The sample injection volume was 50 μL. The standard hydroxy acid form of Lv was prepared from its lactone form (Sigma-Aldrich, St. Louis, MO, USA) according to Nyilasi et al. [34]. Finally, lactone and hydroxy acid forms were calculated as the Lv yield.

### 2.5. Identification of Organic Compounds from Post-Fermented Substrates by Gas Chromatography-Mass Spectrometry

As described above, the samples (1 mL) from the post-fermented oat straw and the post-fermented wheat bran with CDBB H-194 were obtained [35]. Afterwards, they were derivatized with 60 µL of pyridine and 160 µL of N, O-Bis(trimethylsilyl) trifluoro acetamide with trimethylchlorosilane (BSTFA, 1% TMS). They were injected into the Clarus 580 GC–MS instrument (Perkin Elmer, Waltham, MA, USA) coupled to Clarus SQ. 8S MS (Perkin Elmer, Waltham, MA, USA) with Elite-5 MS column (30 m × 0.32 mm × 0.25 μm). Helium gas was used at a constant flow rate of 0.55 mL/min. The oven temperature program was set to an initial temperature of 80 °C/3 min, a ramp to 180 °C by 5 °C steps, then 10 °C/min to 280 °C held for 10 min. The injector port temperature was 250 °C. Inlet line temperature of 200 °C, source temperature of 230 °C, solvent delay of 3 min and Mass of 30–500 m/z. Compounds were identified by comparing their mass spectral fragmentation with the library of the National Institute of Standards and Technology (NIST).

### 2.6. Calculations and Statistical Analysis

Data of Lv yield were analyzed in a 2^2^ factorial design (two substrates and two strains of *A. terreus*) with repeated measurements [36]. The concentration of Lv was the main response variable (dependent variable), whereas the strains, substrates, and fermentation time were treated as fixed effects. Statistical processing was based on the SAS PROC MIXED procedure [37]. The covariance structure was selected regarding the Information Criterion of Akaike and Bayesian Criteria of Schwarz [38].

Final concentrations of Lv and zero-order kinetic coefficients of Lv were analyzed as a 2^2^ factorial experiment (Appendix A document). The test of Tukey was used for means comparison [36].

Degradation efficiencies of lignin, ‘cellulose + hemicellulose’, and an ad hoc index *ε* were determined. Also, the results were analyzed regarding the efficiency of SSF (*E_SSF_*) which is the lignin degradation compared to ‘cellulose + hemicellulose’ breakdown [39].

The degradation efficiencies of lignin and ‘cellulose + hemicellulose’ in oat straw and wheat bran were estimated by Equations (1) and (2), respectively. The detailed derivation of Equations (1) and (2) is included in Appendix A document.
*η*_lig_ = 1 − (*γ*_ligf_/*γ*_ligi_) × [(1 − *γ*_ligi_)/(1 − *γ*_ligf_)](1)
*η*_(c + h)_ = 1 − (*γ*_(c + h)__f_/*γ*_(c + h)__i_) × [(1 − *γ*_(c + h)__i_)/(1 − *γ*_(c + h)__f_)](2)
where *γ*_ligi_ and *γ*_(c + h)i_ are the lignin and ‘cellulose + hemicellulose’ contents in the material, respectively. γ_ligf_ and *γ*_(c + h)f_ are the final contents of lignin and ‘cellulose + hemicellulose’ in the substrate, respectively. All γ are in kg/kg of DM Equations (2) and (3) are in decimals; they can be multiplied by 100 to report the removal efficiencies in %.

We defined an ad hoc indicator (*ε*) according to Equation (3)
*ε = η*_lig_/*η*_(c + h)_(3)
where *η*_lig_ and *η*_(c + h)_ are degradation efficiencies of lignin and ‘cellulose + hemicellulose’ in oat straw and wheat bran, respectively.

This parameter would indicate whether ‘cellulose + hemicellulose’ degradation was higher than lignin. For instance, *ε <* 1.0 would indicate efficiency of degradation of ‘cellulose + hemicellulose’ higher than that of lignin.

The *E_SSF_* was calculated with Equation (4) below:*E_SSF_* (%) = [(loss of lignin)/(loss of hemicellulose + cellulose)] × 100(4)

## 3. Results and Discussion

The chemical composition of substrates for SSF is shown in Table 2. As expected, wheat bran had the highest CP and EE content of both substrates, whereas oat straw contained the highest lignocellulose concentrations according to NDF and ADF values.

Although wheat bran provided a better nutrient composition, this was not translated into a higher Lv yield. The SSF of oat straw using CDBB H-194 strain at 16 d of incubation time gave the highest Lv final concentration (23.8 mg/g DM _fed_) (*p* < 0.001; Figure 1a). However, Lv yield with the same strain and wheat bran was three-fold lower than oat straw (6.8 mg/g DM _fed_; Figure 1a). An average slope of 1.4 mg Lv/g DM _fed_ was observed during treatment with CDBB H-194 and oat straw, even on the last day of incubation (Appendix A document). This trend strongly suggested that it is likely to obtain Lv concentrations higher than 23.8 mg/g DM _fed_ if the incubation time was extended.

Gulyamova et al. [11] studied the SSF of oat bran (the closest substrate to oat straw reported in the open literature) using two strains of *A. terreus* (20 and 40). They obtained moderate Lv yields of 9.54 y 8.4 mg/g DM _fed_, respectively (Table 1).

Regarding the works dealing with SSF of wheat-based substrates for Lv production, Pansuriya et al. [10] obtained a low yield of Lv 3.7 mg/g DM with the strain *A. terreus* UV 1718 using wheat bran, whereas Patil et al. [8] reported an Lv yield of 12.5 mg/g DM for an *A. terreus* strain. More recently, Bashir et al. [13] evaluated the SSF of wheat straw for Lv production by *A. terreus* in various operational conditions. They reported that the highest concentration of Lv obtained was 60 mg/g DM (Table 1). Unfortunately, the article did not provide information on strain availability, so the reproducibility of such results is debatable. The article also lacks information on substrate characterization, substrate degradation efficiencies, kinetic of Lv production, and characterization of organic compounds present in the post-fermented materials (Table 1).

Lovastatin yield using oat straw and CDBB H-194 strain was on the high side of results compiled in Table 1. This could be due to some composition features of SSF media used in this work. For instance, the use of glycerol in SSF could positively influence Lv yields. Zhang et al. [40] reported that the use of absorbent compounds such as glycerol makes the SSF process more robust and resistant to catabolite repression. They indicated that glycerol addition could lead to a glycerol concentration gradient; this could have generated an appropriate environment for fungal growth and the colonization of lignocellulolytic substrates.

The addition of methionine in our culture medium is another possible explication for the rising yield of Lv. It is known that dihydromonacolin L is an intermediate in the Lv biosynthesis; the latter possesses two methyl groups (C-2 and C-6 positions), the C-6 is derived from methionine [41]. This view is supported by experiments by Rollini and Manzoni [42], who evaluated the Lv production with *A. terreus* and methionine (0.1 g/L) in the culture medium. Their results suggested that methionine added in the culture medium yielded 20% higher Lv than the control medium. To some extent, our results agreed with these previous findings.

In this study, SSF by CDBB H-1976 strain resulted in the second highest Lv yield (9.1 mg/g DM _fed_) with oat straw as substrate at 12 d incubation (Figure 1b). For the CDBB H-1976 strain with wheat bran, the maximum Lv yield was 5.67 mg/g DM _fed_ at 16 d (Figure 1b). Prior studies have noted the importance of genes and enzymes in Lv biosynthesis [43]. In this context, Praveen et al. [44] demonstrated that *A. terreus* isolated from soil was an Lv producer, while an endophytic fungus of the same genus could not produce detectable levels of this compound. Likewise, Bhargavi et al. [45] indicated that the homology with the Lv gene cluster of an endophytic *A. terreus* was much lower than an *A. terreus* isolated from soil. In this work, *A. terreus* CDBB H-194 was isolated from the soil, while CDBB H-1976 was isolated from patients with pulmonary aspergillosis [46]. A possible explanation for lower Lv yield by CDBB H-1976 could be the possible absence of Lv coding genes in this strain, rather than physiological and environmental factors [47].

An analysis of variance of the 2^2^ factorial experiment considering only the final concentrations of Lv at 16-d incubation indicated a significant interaction between the strain and substrate factors (*p* < 0.0001; Appendix A document). Indeed, the means test confirmed that the Lv yield of treatment *A. terreus* CDBB H-194 with oat straw was significantly superior (Appendix A document).

The coefficient of determination R^2^ in regressions of the zero-order kinetic model adequately fitted the experimental data of Lv concentrations (R^2^ between 0.8453–0.9914). Regression significances were very high, as revealed by very low *p*-values (range 0.0000000125 to 0.0029) [48]. The zero-order kinetics of Lv concentration increase was consistent with remarks by Levenspiel [48], who observed that the zero-order model often fits very well with the kinetic patterns in systems where the initial concentration of substrate is very high (in the present report, ‘cellulose + hemicellulose’ contents of substrates were very high, Table 3).

The values for the slopes in the linear regression of Lv concentration showed that the trend of the rate coefficients was paralleled with the trends of Lv final concentrations in the four treatments. A strong interaction between the experimental factors such as that of the final concentration of Lv was also observed for the response variable slope *b_1_* (*p* < 0.0004; Appendix A document). The test of means also confirmed that the kinetic coefficient *b_1_* of the treatment *A. terreus* CDBB H194/oat straw was significantly higher than the other three treatments (Appendix A document).

The lignocellulose-degrading efficiency and the identification of organic compounds from post-fermented substrates by GC–MS analyses were based upon the treatments using *A. terreus* CDBB H-194 due to the highest Lv yield. The results show that the degradation efficiencies of ‘hemicellulose + cellulose’ *η*_(c + h)_ in oat straw (24.1%) and wheat bran (30.2%) were higher than the efficiency degradation of lignin *η*_lig_, which was approximately 15.0% for both substrates (Table 3). Additionally, the decrease in hemicellulose and cellulose in samples of wheat bran resulted in being statistically significant, with *p* < 0.040 and *p* < 0.004, respectively (Table 3). These values support the concept of the SSF bioprocess with yeast or fungi resulting in a decrease in the hemicellulose and cellulose from the agricultural residues [49,50,51].

The *ε* indicator for oat straw (0.6191) and wheat bran (0.5268) also confirmed that SSF of both substrates by CDBB H-194 consumed more ‘hemicellulose + cellulose’ than lignin. These results are consistent with those of Darwish et al. [49] who performed an investigation on upgrading the nutritional value of maize stalks using SSF with white rot fungus *Pleurotus ostreatu*; they found *η*_lig_, *η*_(c + h)_, and *ε* values of 11.9%, 17.0%, and 0.705, respectively at 14 d of incubation.

The value *E_SSF_* _of_ 14.24% in oat straw and 9.38% in wheat brand fermentation indicated that 0.14 g and 0.09 g of lignin were degraded per 1 g of ‘hemicellulose + cellulose’ consumed at 16 d, respectively (Table 3). This result was congruent with our values of the degradation efficiencies and indicators *ε* discussed above. It was somewhat expected because *A. terreus* has a wide array of enzymes to degrade ‘cellulose and hemicellulose’, but the fungal strain is not a lignin degrader [52]. In addition, the findings of the current work were consistent with those of Azlan et al. [12], who showed that the degradation of ‘cellulose + hemicellulose’ was 20.5% by *A. terreus* using rice straw as the substrate.

The organic compounds profile from post-fermented oat straw with *A. terreus* CDBB H-194 is shown in Table 4 and the chromatogram in Figure 2a. A total of 23 compounds were identified as follows (we merged the repetitions):

*1 statin and 3 polyols:* simvastatin, trimethylsilyl ether of glycerol; xylitol, 1,2,3,4,5-pentakis-o-(trimethylsilyl)-; 1-monooleoylglycerol trimethylsilyl ether; *15 organic acids:* dodecanedioic acid, 3,6-dioxa-2,7-disilaoctane, 2,2,4,7,7-pentamethyl-; silane, [(11-bromoundecyl)oxy]trimethyl-; butanoic acid, 3-methyl-2-[(trimethylsilyl)oxy]-, trimethylsilyl ester; gulonic acid, 2,3,5,6-tetrakis-o-(trimethylsilyl)-, lactone; propanoic acid, 2-[(trimethylsilyl)oxy]-, trimethylsilyl ester; hexacosanoic acid, methyl ester; heptadecanoic acid, trimethylsilyl ester; 9-octadecenoic acid (z)-, methyl ester; linoleic acid ethyl ester; ethyl 9-hexadecenoate; 9,12-octadecadienoic acid (z,z)-, trimethylsilyl ester; oleic acid, trimethylsilyl ester; cholesterol trimethylsilyl ether; 1,3-dipalmitin trimethylsilyl ether; *4 others*: bis [2-trimethylsiloxy]ethyl sulfone; 3á,4á-bis(trimethylsiloxy)cholest-5-ene; cystathionine, bis(triemthylsilyl) ester; 3beta-hydroxy-5-cholen-24-oic acid; bis(trimethylsilyl) ester.

As shown in Table 4, 9 out of 30 compounds exhibited areas larger than 2%.

The GC–MS analysis confirmed the presence of simvastatin, a semi-synthetic statin derived from the alkylation of Lv [17]. It was reported that *A. terreus* could produce Lv with the concomitant formation of simvastatin as a secondary metabolite in submerged fermentation experiments of oat grain [11]. A remarkable feature of simvastatin (10 mg/L culture media) is that it reduced the in vitro methane production (*p* < 0.05) of a diet containing 70% forage and 30% concentrate as a substrate and rumen fluid inoculum from Holstein cows [53]. Thus, the presence of simvastatin in the post-fermented oat straw could enhance its contribution to methane mitigation if used as a supplement for ruminants.

The organic compounds obtained from the post-fermented wheat bran are presented in Table 5 and the corresponding chromatogram in Figure 2b. A total of 14 compounds were identified and grouped (we merged the repetitions), including *2 polyols:* trimethylsilyl ether of glycerol; xylitol, 1,2,3,4,5-pentakis-o-(trimethylsilyl)-, *2 sterol esters*: carbonic dihydrazide; stearic acid hydrazide; *8 organic acids*: dodecanedioic acid, silane, [(11-bromoundecyl)oxy]trimethyl-; gulonic acid, 2,3,5,6-tetrakis-o-(trimethylsilyl)-, lactone; d-mannitol, 1,2,3,4,5,6-hexakis-o-(trimethylsilyl)-; 9,12-octadecadienoic acid (z,z)-, trimethylsilyl ester; 9-octadecenoic acid (z)-, methyl ester; 8,11,14-eicosatrienoic acid, (z,z,z)-; 1-monooleoylglycerol trimethylsilyl ether), and *three others* (bis(trimethylsilyl) ester; guanosine; carotene. Overall, 10 compounds out of 22 exhibited areas larger than 2% (Table 5).

The present findings of some lipids and lipid derivatives in the post-fermented wheat bran (e.g., 9-octadecenoic acid, 9,12-octadecadienoic acid, and dodecanoic acid) were consistent with those of del Río et al. [54], who characterized the lipids in wheat straw by GC–MS analyses.

Finally, only a couple of possible antimicrobial compounds were detected: a small amount of the antifungal agent 1,3-dipalmitin trimethylsilyl ether (area 1.14%) was identified in the post-fermented oat straw [55], whereas the antimicrobial agent, stearic acid hydrazide (area 13%), was identified in the post-fermented wheat bran [56].

## 4. Conclusions

Comparing two Aspergillus strains revealed that the CDBB H-194 strain using oat straw as substrate exhibited the highest potential to produce Lv (23.8 mg/g DM fed). The experiment showed a significant interaction between the strain and substrate factors. Lovastatin concentrations increase adequately fitted a zero-order kinetic model for the whole-time course of treatment CDBB H-194 strain/oat straw and the early stages of the fermentation of the other three treatments.

GC–MS analysis confirmed the presence of simvastatin in the post-fermented oat straw, which could enhance its contribution to rumen methane mitigation. Furthermore, only a couple of compounds were identified that could negatively affect ruminal bacteria and fungi: (i) 1,3-dipalmitin trimethylsilyl ether was found in the post-fermented oat straw and (ii) stearic acid hydrazide was observed in the post-fermented wheat bran, using *A. terreus* CDBB H-194 in both cases.

Finally, due to its high yield of Lv from post-fermented oat straw with CDBB H-194, low dietary proportions could be used as an Lv-carrier supplement for rumen methane mitigation in further research.

## Figures and Tables

**Figure 1 animals-12-02697-f001:**
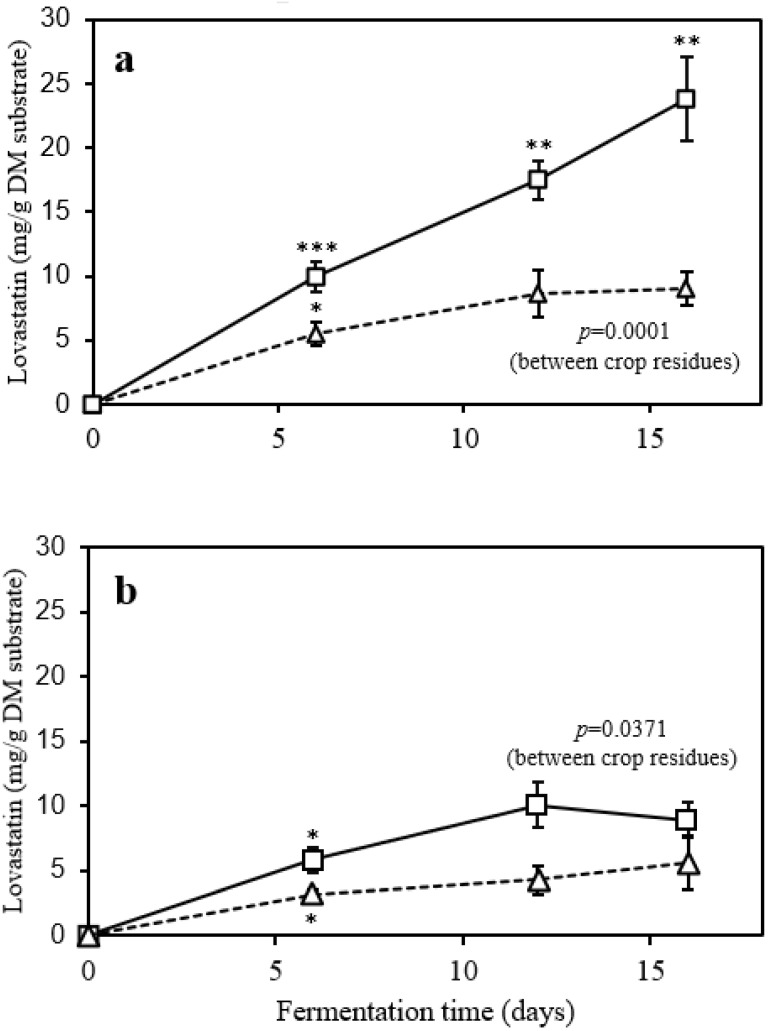
Lovastatin production by SSF: (**a**) *A. terreus* CDBB H-194 and (**b**) *A. terreus* CDBB H-1976 with two crop residues as substrates. Keys: ☐, oat straw; △, wheat bran. Data represent the mean ± standard deviation. Asterisks in lines indicate significant differences from repeated measures as determined by the procedure of Tukey (* *p* < 0.05; ** *p* < 0.01; *** *p* < 0.0001).

**Figure 2 animals-12-02697-f002:**
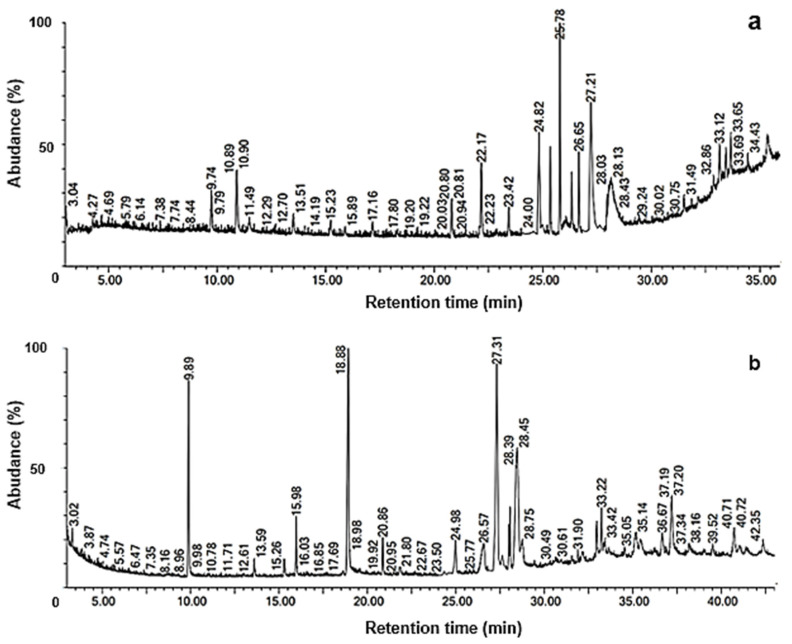
Chromatograms of GC–MS profile from post-fermented substrates (**a**) oat straw and (**b**) wheat brand.

**Table 1 animals-12-02697-t001:** Lovastatin production using *Aspergillus* strains in solid-state fermentation of agricultural residues.

SSF Process	Performance and Results	Remarks	Ref.
Strain: *A. terreus* PM3 Spore concentration: 10^8^ spores/mL Substrate: wheat bran	T = 25 °C, M_0_ = 60%, Fermentation time = 10 d, pH = 7.1. [Lv] = 12.5 mg/g DM substrate	Sa: No/0; Bcs: No/0; Des: No/0; Sar: No/0; Klp: No/0; Coc: No/0 Score 0.	[8]
Strain: *A. terreus* MTCC 279 Spore concentration: 10^6^ spores/mL Substrate: wheat bran.	T = 25 °C, M_0_ = 66%, Fermentation time = 3 d, pH = 6, [Lv] = 13.4 mg/g DM substrate	Sa: Yes/1; Pas: No/0; Des: No/0, Sar: Yes/1; Klp: No/0; Coc: No/0 Score 2.	[9]
Strain: *A. terreus* UV 1718 Spore concentration: 10^8^ spores/mL Substrate: wheat bran	T = 28 °C, M_0_ = 70%, Fermentation time = 10 d, pH = 6, [Lv] = 3.7 mg/g DM substrate	Sa: No/0; Pas: No/0; Des: No/0; Sar: Yes/1; Klp: No/0; Coc: No/0 Score 1.	[10]
Strain: *A. terreus* 20 Spore concentration: 10^7^ spores/mL Substrate: Oat bran	T = 28 °C, M_0_ = 55–65%, Fermentation time = 14 d, pH = 7.5, [Lv] = 9.5 mg/g DM substrate	Sa: No/0; Pas: No/0; Des: No/0; Sar: No/0; Klp: No/0; Coc: No/0 Score 0.	[11]
Strain: *A. terreus* ATCC 74135 Spore concentration: 10^7^ spores/mL Substrate: rice straw	T = 25 °C, M_0_ = 50%, Fermentation time = 14 d, pH = 6, [Lv] = 0.69 mg/g DM substrate	Sa: Yes/1; Pas: Yes/1; Des: No/0; Sar: Yes/1; Klp: No/0; Coc: No/0 Score 3.	[12]
Strain: *A. terreus* Spore concentration: 10^8^ spores/mL Substrate: wheat straw	T = 30 °C, M_0_ = Not reported, Fermentation time = 8 d, pH = 7.3, [Lv] = 60 mg/g DM substrate	Sa: No/0; Pas: No/0; Des: No/0; Sar: Yes/1; Klp: No. Coc: No/0 Score 1.	[13]
Strain: *A. terreus* FFCBP-1053 Substrate: rice straw Spore concentration: 5 × 10^7^–5 × 10^8^ spores/mL	T = 35 °C, M_0_ = 70%, Fermentation time = 10 d, pH = 4.5, [Lv] = 2.1 mg/g DM substrate	Sa: Yes/1; Pas: No/0; Des: No/0; Sar: Yes/1; Klp: No/0; Coc: No/0 Score 2.	[14]
Strain: *A terreus* CDBB H-194 Substrate: Oat straw Spore concentration: 10^7^ spores/mL	T = 26 °C, M_0_ = 70%, Fermentation time = 16 d, pH = 5.5, [Lv] = 23.8 mg/g DM substrate	Sa: Yes/1; Pas: Yes/1; Des: Yes/1; Sar: Yes/1; Klp: Yes/1; Coc: Yes/1 Score 6.	This work

T: fermentation temperature; M_0:_ initial moisture, [Lv]: Sa: strain availability, Pas: proximate analysis of substrates, Des: degradation efficiency of substrates, Sar: statistical analysis of results, Klp: kinetics of Lv production. Coc: characterization of organic compounds from post-fermented substrates.

**Table 2 animals-12-02697-t002:** Proximate analysis of substrates for solid-state fermentation.

Item	Oat Straw	Wheat Bran
Dry Matter (g/kg)	945.2 ± 25.1	932.6 ± 18.7
Chemical composition parameter (g/kg DM):		
CP ^a^	43.3 ± 6.1	151.8 ± 12.0
EE ^b^	31.1 ± 4.8	47.5 ± 5.2
Ash	62.6 ± 5.5	77.5 ± 6.7
NDF ^c^	680.2 ± 52.6	568.0 ± 47.3
ADF ^d^	415.8 ± 27.3	174.2 ± 15.8

^a^ Crude protein, ^b^ ether extract, ^c^ neutral detergent fiber, ^d^ acid detergent fiber. Data represent the mean ± standard deviation of triplicates.

**Table 3 animals-12-02697-t003:** Cell wall composition and degradation efficiencies of lignin and ‘cellulose + hemicellulose’ of SSF by *A. terreus* CDBB H-194 at 16 d.

	Oat Straw		Wheat Bran	
Item	Unfermented	Fermented	Unfermented	Fermented
Hemicellulose (g/kg DM)	267.5 ± 12.6	232.1 ± 15.5	393.9 ± 18.3	340.2 ± 16.0 * ^e^
Cellulose (g/kg DM)	324.2 ±11.2	291.7 ± 12.0	118.0 ± 8.3	82.2 ± 7.4 * ^f^
Lignin (g/kg DM)	83.8 ± 10.2	74.2 ± 5.18	56.0 ± 3.0	47.6 ± 4.2
*η*_(c + h)_ ^a^ (%)	–	24.10	–	30.24
*η*_lig_ ^b^ (%)	–	14.92	–	15.93
*ε* (-) ^c^	–	0.6191	–	0.5268
*E_SSF_* (%) ^d^	–	14.14	–	9.38

^a^ Degradation efficiency of cellulose + hemicellulose, ^b^ degradation efficiency of lignin, ^c^ ratio of *η*_lig_ to *η*_(c + h)_, ^d^ efficiency of SSF according to Equation (4). Data represent the mean ± standard deviation of triplicates. The asterisk (*) denotes statistically differences within groups as determined by Student’s *t* tests (*p* < 0.040); specifically, ^e^ *p* < 0.040 and ^f^ *p* < 0.004.

**Table 4 animals-12-02697-t004:** Identification of organic compounds from post-fermented oat straw by GC–MS.

PN ^a^	RT ^b^ (Min)	Compound Name	# CAS	Area (%)
1	3.319	3,6-Dioxa-2,7-disilaoctane, 2,2,4,7,7-pentamethyl-	17887-27-3	0.509
2	9.891	Trimethylsilyl ether of glycerol	6787-10-6	6.020
3	13.593	Silane, [(11-bromoundecyl)oxy]trimethyl-	26305-83-9	0.631
4	15.303	Butanoic acid, 3-methyl-2-[(trimethylsilyl)oxy]-, trimethylsilyl ester	55124-92-0	0.621
5	15.979	Gulonic acid, 2,3,5,6-tetrakis-O-(trimethylsilyl)-, lactone	55528-75-1	1.648
6	18.915	bis [2-Trimethylsiloxy]ethyl sulfone	97916-04-6	11.670
7	20.861	Xylitol, 1,2,3,4,5-pentakis-O-(trimethylsilyl)-	14199-72-5	1.136
8	21.811	Propanoic acid, 2-[(trimethylsilyl)oxy]-, trimethylsilyl ester	17596-96-2	0.458
9	24.982	Hexacosanoic acid, methyl ester	5802-82-4	1.787
10	26.575	Heptadecanoic acid, trimethylsilyl ester	55517-58-3	3.270
11	27.313	9-Octadecenoic acid (Z)-, methyl ester	112-62-9	14.416
12	27.987	Linoleic acid ethyl ester	544-35-4	0.867
13	28.062	Ethyl 9-hexadecenoate	54546-22-4	1.271
14	28.447	9,12-Octadecadienoic acid (Z,Z)-, trimethylsilyl ester	56259-07-5	12.062
15	28.747	Oleic acid, trimethylsilyl ester	21556-26-3	2.250
16	32.134	Dodecanedioic acid, bis(trimethylsilyl) ester	22396-19-6	0.601
17	32.959	1-Monooleoylglycerol trimethylsilyl ether	54284-47-8	1.492
18	33.224	1-Monooleoylglycerol trimethylsilyl ether	54284-47-8	2.152
19	33.344	3á,4á-Bis(trimethylsiloxy)cholest-5-ene	33287-25-1	0.492
20	33.419	1-Monooleoylglycerol trimethylsilyl ether	54284-47-8	0.743
21	33.634	Dodecanedioic acid, bis(trimethylsilyl) ester	22396-19-6	0.630
22	35.180	Cholesterol trimethylsilyl ether	1856-05-9	2.149
23	35.415	9,12-Octadecadienoic acid (Z,Z)-, trimethylsilyl ester	56259-07-5	1.560
24	36.666	3Beta-hydroxy-5-cholen-24-oic acid	5255-17-4	1.243
25	37.196	Simvastatin	79902-63-9	4.141
26	38.156	Cystathionine, bis(triemthylsilyl) ester	73090-79-6	0.533
27	39.522	Dodecanedioic acid, bis(trimethylsilyl) ester	22396-19-6	0.470
28	40.722	Cholesterol trimethylsilyl ether	1856-05-9	1.822
29	41.027	Dodecanedioic acid, bis(trimethylsilyl) ester	22396-19-6	0.645
30	42.358	1,3-Dipalmitin trimethylsilyl ether	53212-95-6	1.148

^a^ Peak number, ^b^ retention time.

**Table 5 animals-12-02697-t005:** Identification of organic compounds from post-fermented wheat bran by GC–MS.

PN ^a^	RT ^b^ (Min)	Compound Name	# CAS	Area (%)
1	9.741	Trimethylsilyl ether of glycerol	6787-10-6	1.753
2	10.901	Silane, [(1-methyl-1,3-propanediyl)bis(oxy)]bis[trimethyl-	56771-47-2	2.963
3	11.492	Carbonic dihydrazide	497-18-7	0.686
4	13.508	Stearic acid hydrazide	4130-54-5	13.508
5	15.233	Carbonic dihydrazide	497-18-7	0.66
6	20.81	Xylitol, 1,2,3,4,5-pentakis-O-(trimethylsilyl)-	14199-72-5	1.635
7	22.166	Gulonic acid, 2,3,5,6-tetrakis-O-(trimethylsilyl)-, lactone	55528-75-1	3.484
8	23.421	Dodecanedioic acid, bis(trimethylsilyl) ester	22396-19-6	1.005
9	24.819	D-Mannitol, 1,2,3,4,5,6-hexakis-O-(trimethylsilyl)-	14317-07-8	5.914
10	25.339	Gulonic acid, 2,3,5,6-tetrakis-O-(trimethylsilyl)-, lactone	55528-75-1	2.513
11	25.784	Xylitol, 1,2,3,4,5-pentakis-O-(trimethylsilyl)-	14199-72-5	5.427
12	26.329	Dodecanedioic acid, bis(trimethylsilyl) ester	22396-19-6	1.443
13	26.655	Gulonic acid, 2,3,5,6-tetrakis-O-(trimethylsilyl)-, lactone	55528-75-1	2.171
14	27.211	9,12-Octadecadienoic acid (Z,Z)-, methyl ester	112-63-0	10.538
15	27.952	9-Octadecynoic acid, methyl ester	1120-32-7	0.99
16	28.132	8,11,14-Eicosatrienoic acid, (Z,Z,Z)-	1783-84-2	10.041
17	31.493	Guanosine	118-00-3	0.632
18	32.859	1-Monooleoylglycerol trimethylsilyl ether	54284-47-8	0.662
19	33.124	1-Monooleoylglycerol trimethylsilyl ether	54284-47-8	1.333
20	33.424	9,12-Octadecadienoic acid (Z,Z)-, trimethylsilyl ester	56259-07-5	1.011
21	33.649	8,11,14-Eicosatrienoic acid, (Z,Z,Z)-	1783-84-2	1.541
22	35.335	á Carotene	7235-40-7	2.235

^a^ Peak number, ^b^ retention time.

## Data Availability

All data supporting our findings are included in the manuscript and the Appendix A document. Selected Excel files are available from the authors upon request. The *Aspergillus* strains used in this study are available from the National Collection of Microbial and Cellular Cultures of the CINVESTAV-IPN.

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
