# Peer review of "Bioprocessing of Two Crop Residues for Animal Feeding into a High-Yield Lovastatin Feed Supplement"

_animals, 2022, doi:10.3390/ani12192697_

Round 1
Reviewer 1 Report
This study investigates significant potential solutions for mitigation of methane production in ruminants. Further investigation for Solid State Fermentation of Oat Straw with the strain CDBB H-194 could show more promising results and authors should conduct further research in this direction.
Author Response
Please note that we could upload here the Revised MS and the Revised Conveying Letter.
Below we describe the objectives and structure of the Response-to-Reviewers document (RtR). The answers to your Comments are included in the RtR document,
The Authors wish to convey their sincere gratitude to you, the Editors and the other Reviewers for the management of the article and the careful reading of our paper Animals-1924907 (Bioprocessing of two crop residues for animal feeding into a high-yield lovastatin feed supplement).
Several recommendations and suggestions of the Editor and the Reviewers were insightful, very inspiring, and allowed to significantly improve the manuscript.
We are also very grateful for the deadline extension.
Regarding the first file, the Response to Reviewers document (RtR, in short), we would like to share this information:
-we have made a serious effort to answer, one by
one, all the comments and suggestions raised by the Editor and the Reviewers.
-In section B of the RtR, all the original comments have been rewritten and an ordinal number was assigned. This Comment number corresponds to the same Comment that appears in the document. The number of the Comment is ours, but the Comment is always an original Comment of one or more of the Reviewers.
-Each Comment is organized as follows:
First, the comment/question/suggestion of the Reviewer is written verbatim as an easy access reminder of the Reviewer’s remark
Second, we created the subsection ‘Answer’ where, we communicate our decision to modify or defend, we develop the discussion of the issue and often we might include new working tables or figures to better explain our discussion.
Third, we created the subsection ‘Before it read’. This subsection contains an excerpt of the original MS that will be modified, verbatim
Fourth, we created the subsection ‘Now it reads’. This subsection contains the modified passage (either line, paragraph, table, or figure, etc.) in RED color that will be placed in the RMS.
-We have thoroughly revised the English language in the revised article, as requested by the Editor and two Reviewers.
First, we revised the manuscript using a free version of Grammarly Premium.
Second, the preliminary revised article was further checked for English syntax and usage by two co-authors who pursued their doctoral studies in England.
Third, we checked the revised article using Grammarly Premium, the complete version of Grammarly.
The evaluation of the revised article made by Grammarly showed that the score of the revised article increased from 96% in the penultimate version to 98% in the last version of the manuscript. The missing 2% is constituted by Grammarly recommendation/mistakes on the section of the References list of the MS. However, this section was not modified since Grammarly recommended changes in the titles of the articles, changes to last names of authors, changes to the DOI. Changing this evidently was a No-No.
Therefore, the Grammarly score of the RMS (text) is 100%, disregarding the References list at the end of the article.

Reviewer 2 Report
The article entitled “Bioprocessing of Two Crop Residues for Animal Feeding into a High-yield Lovastatin Feed Supplement” is a good initiative to use bioprocessing of crop residues from an economic standpoint. It is well organized and the English are good; only a few articles are missing. The manuscript is meticulously prepared; however, a few points I would like to add to improve its content to be published.
In Materials & method
In section 2.3. Conditions of solid-state fermentation -Five grams of the substrate was ground (5 mm), -
Please write down the substrate name. Are they ‘Oat straw’ and ‘Wheat bran’?
In section, 2.4 Post-fermented substrates….
Please mention the substrate's name (Oat straw or Wheat bran) also.
In Result
In Table 2. The author mentioned “Proximate analysis of substrates for solid-state fermentation”.
It would be great if the author could add another table for all proximate analysis results of the two crop residues before and after bioprocessing.
In Table 3. Only cell wall composition has been compared. However, it needs to be mentioned the p values of the data. Otherwise, it is difficult to understand if the bioprocessing of crop residues has a significant effect on the chemical composition of the feed supplement.
In Discussion
Please mention in the discussion any changes in the result section with references.
Author Response
Please note that we could upload here the Revised MS and the Revised Conveying Letter.
Below we describe the objectives and structure of the Response-to-Reviewers document (RtR). Our answers to your Comments are included in the RtR document.
The Authors wish to convey their sincere gratitude to you, the Editors and the other Reviewers for the management of the article and the careful reading of our paper Animals-1924907 (Bioprocessing of two crop residues for animal feeding into a high-yield lovastatin feed supplement).
Several recommendations and suggestions of the Editor and the Reviewers were insightful, very inspiring, and allowed to significantly improve the manuscript.
We are also very grateful for the deadline extension.
Regarding the first file, the Response to Reviewers document (RtR, in short), we would like to share this information:
-we have made a serious effort to answer, one by
one, all the comments and suggestions raised by the Editor and the Reviewers.
-In section B of the RtR, all the original comments have been rewritten and an ordinal number was assigned. This Comment number corresponds to the same Comment that appears in the document. The number of the Comment is ours, but the Comment is always an original Comment of one or more of the Reviewers.
-Each Comment is organized as follows:
First, the comment/question/suggestion of the Reviewer is written verbatim as an easy access reminder of the Reviewer’s remark
Second, we created the subsection ‘Answer’ where, we communicate our decision to modify or defend, we develop the discussion of the issue and often we might include new working tables or figures to better explain our discussion.
Third, we created the subsection ‘Before it read’. This subsection contains an excerpt of the original MS that will be modified, verbatim
Fourth, we created the subsection ‘Now it reads’. This subsection contains the modified passage (either line, paragraph, table, or figure, etc.) in RED color that will be placed in the RMS.
-We have thoroughly revised the English language in the revised article, as requested by the Editor and two Reviewers.
First, we revised the manuscript using a free version of Grammarly Premium.
Second, the preliminary revised article was further checked for English syntax and usage by two co-authors who pursued their doctoral studies in England.
Third, we checked the revised article using Grammarly Premium, the complete version of Grammarly.
The evaluation of the revised article made by Grammarly showed that the score of the revised article increased from 96% in the penultimate version to 98% in the last version of the manuscript. The missing 2% is constituted by Grammarly recommendation/mistakes on the section of the References list of the MS. However, this section was not modified since Grammarly recommended changes in the titles of the articles, changes to last names of authors, changes to the DOI. Changing this evidently was a No-No.
Therefore, the Grammarly score of the RMS (text) is 100%, disregarding the References list at the end of the article.
Thank you again for your insightful comments.

Reviewer 3 Report
A very interesting contribution to the efforts to reduce methane emissions in ruminants. There are some minor spells, like propor-tions, etc
Author Response
Please note that we could upload here the Revised MS and the Revised Conveying Letter.
Below we describe the objectives and structure of the Response-to-Reviewers document (RtR). Our answers to your Comments are included in the RtR document.
The Authors wish to convey their sincere gratitude to you, the Editors and the other Reviewers for the management of the article and the careful reading of our paper Animals-1924907 (Bioprocessing of two crop residues for animal feeding into a high-yield lovastatin feed supplement).
Several recommendations and suggestions of the Editor and the Reviewers were insightful, very inspiring, and allowed to significantly improve the manuscript.
We are also very grateful for the deadline extension.
Regarding the first file, the Response to Reviewers document (RtR, in short), we would like to share this information:
-we have made a serious effort to answer, one by
one, all the comments and suggestions raised by the Editor and the Reviewers.
-In section B of the RtR, all the original comments have been rewritten and an ordinal number was assigned. This Comment number corresponds to the same Comment that appears in the document. The number of the Comment is ours, but the Comment is always an original Comment of one or more of the Reviewers.
-Each Comment is organized as follows:
First, the comment/question/suggestion of the Reviewer is written verbatim as an easy access reminder of the Reviewer’s remark
Second, we created the subsection ‘Answer’ where, we communicate our decision to modify or defend, we develop the discussion of the issue and often we might include new working tables or figures to better explain our discussion.
Third, we created the subsection ‘Before it read’. This subsection contains an excerpt of the original MS that will be modified, verbatim
Fourth, we created the subsection ‘Now it reads’. This subsection contains the modified passage (either line, paragraph, table, or figure, etc.) in RED color that will be placed in the RMS.
-We have thoroughly revised the English language in the revised article, as requested by the Editor and two Reviewers.
First, we revised the manuscript using a free version of Grammarly Premium.
Second, the preliminary revised article was further checked for English syntax and usage by two co-authors who pursued their doctoral studies in England.
Third, we checked the revised article using Grammarly Premium, the complete version of Grammarly.
The evaluation of the revised article made by Grammarly showed that the score of the revised article increased from 96% in the penultimate version to 98% in the last version of the manuscript. The missing 2% is constituted by Grammarly recommendation/mistakes on the section of the References list of the MS. However, this section was not modified since Grammarly recommended changes in the titles of the articles, changes to last names of authors, changes to the DOI. Changing this evidently was a No-No.
Therefore, the Grammarly score of the RMS (text) is 100%, disregarding the References list at the end of the article.
Thank you again for your contribution.
